# Dimethyl Sulfoxide: A Bio-Friendly or Bio-Hazard Chemical? The Effect of DMSO in Human Fibroblast-like Synoviocytes

**DOI:** 10.3390/molecules27144472

**Published:** 2022-07-13

**Authors:** Manuel Gallardo-Villagrán, Lucie Paulus, David Yannick Leger, Bruno Therrien, Bertrand Liagre

**Affiliations:** 1Institut de Chimie, Université de Neuchâtel, Avenue de Bellevaux 51, CH-2000 Neuchâtel, Switzerland; villagran@outlook.com (M.G.-V.); bruno.therrien@unine.ch (B.T.); 2Laboratoire PEIRENE UR 22722, Faculté de Pharmacie, Université de Limoges, F-87025 Limoges, France; lucie.paulus@unilim.fr (L.P.); david.leger@unilim.fr (D.Y.L.)

**Keywords:** fibroblast-like synoviocytes, dimethyl sulfoxide, rheumatoid arthritis

## Abstract

The effect of dimethyl sulfoxide (DMSO) in rheumatoid arthritis (RA) human fibroblast-like synoviocytes (FLSs) has been studied on five different samples harvested from the joints (fingers, hands and pelvis) of five women with RA. At high concentrations (>5%), the presence of DMSO induces the cleavage of caspase-3 and PARP-1, two phenomena associated with the cell death mechanism. Even at a 0.5% concentration of DMSO, MTT assays show a strong toxicity after 24 h exposure (≈25% cell death). Therefore, to ensure a minimum impact of DMSO on RA FLSs, our study shows that the concentration of DMSO has to be below 0.05% to be considered safe.

## 1. Introduction

Hydrophilicity is key when designing drugs, since it helps the distribution and ensures a precise concentration. However, sometimes, the physiological solubility is poor or null, and an additional solvent is required to prepare stock solutions. Dimethyl sulfoxide (DMSO) is one of the most common solvents for dissolving drugs that show low solubility in water or physiological media for in vitro and in vivo studies [1,2,3,4,5,6]. This is due to the physicochemical characteristics of this solvent (Table 1). DMSO has a high polarization, is aprotic and possesses apolar groups (Figure 1), which account for its amphipathic nature [7]. DMSO can also pass through cell membranes and displace water, which is why it is commonly used as a cryopreservation agent [8,9,10]. In addition, the use of DMSO has been reported to have beneficial effects in certain diseases, such as gastrointestinal disorders [11], brain edema [12] and schizophrenia [13]. However, it is not prudent to use it lightly, since it presents drawbacks both at a chemical and biological level. At the chemical level, being a compound with a high coordinating capacity, it can lead to changes in the structure of drugs. For example, degradation and/or secondary products are often observed with metal-based compounds, and in such cases, the stability of the compounds in DMSO needs to be addressed methodically [14,15]. At the biological level, adverse effects vary enormously with experimental conditions and cell types. Although concentrations in a range of 0.1–0.5% (*v*/*v*) are usually recommended, there is no single criterion for DMSO concentration. It can even improve cell proliferation in small concentrations. For instance, 1.0% (*v*/*v*) of DMSO is enough to affect lactate dehydrogenase activity [16], more than 0.6% reduces proliferation in HepG2, MDA-MB-231 and MCF-7 cancer cell lines [17], 0.01% increases proliferation in stem cells [18], 0.1% induces alterations in cardiac and hepatic cells [19] and 2% reduces the viability of human blood cells [20]. Given this lack of standardization, we decided to carry out the evaluation of DMSO toxicity in fibroblast-like synoviocytes (FLSs) from patients with rheumatoid arthritis (RA) cells, with which we currently work with and for which there is no defined criteria on the use of DMSO. In this way, we could reliably evaluate the results of our research, confidently ruling out the significant influence of DMSO on cellular viability.

## 2. Results and Discussion

In order to determine the effect of DMSO on FLSs, we decided to use FLSs from five different patients with RA, and from different joints: RA FLSs from the fingers of 60-, 71- and 72-year-old women (samples 60F, 71F and 72F), RA FLSs from the hands of a 65-year-old woman (65H) and RA FLSs from the pelvis of a 69-year-old woman (69P). RA FLSs were isolated from extracted tissue in synovectomy and treated as explained in the experimental part. We first decided to find a suitable DMSO concentration range using only one of the samples (71F). To do this, we first established a wide range of concentrations and evaluated the mortality rate using flow cytometry. Next, we narrowed the range and evaluated the presence of cell death mediators using protein quantification and Western blotting. Finally, once we established a precise range of concentrations to be evaluated, antiproliferative assays (MTT) were carried out on all samples (60F, 65H, 69P, 71F and 72F).

### 2.1. Cell Death Evaluation

First, we performed a flow cytometry study to measure the population of alive/dead cells using a wide range of DMSO concentrations (0, 2.5, 5, 7.5 and 10%) on 71F cells. It should be noted that when using concentrations higher than 5% of DMSO, we observed that most cells were floating in the culture media 24 h after the addition of the DMSO solution. This could be indicative of damage or degradation in the cell membrane as a consequence of initial apoptotic stages and/or the downregulation of adhesion proteins located in the plasmatic membrane [24,25]. Using 7.5 and 10% of DMSO, there were no cells adhering at the bottom of the flask. The results observed in cytometry showed clear signals of high toxicity at high concentrations. While the 2.5% DMSO dose showed seven times more dead cells than the control experiment (0% DMSO), the 5% DMSO dose gave rise to thirty-five times more dead cells, forty times more dead cells with 7.5% and forty-seven times more dead cells with the 10% DMSO concentration (Figure 2). At this point, based on the results, since our main objective was to find adequate and nontoxic concentrations of DMSO in RA FLSs, we reduced the range of the concentrations in the following tests, establishing 5% as the maximum concentration tolerated by RA FLSs.

Subsequently, we evaluated one of the main mediators of programmed cell death, the enzyme caspase-3’s activity [26]. This enzyme makes the cell death process more efficient and its role in apoptosis is considered essential [27]. Caspase-3 remains inactive until it is cleaved once apoptotic signaling processes have occurred [28,29]. Increasing concentrations of DMSO were evaluated (0, 0.05, 0.5, 1, 2.5 and 5% DMSO) in 71F over a 24 h exposure. Only in the highest evaluated concentration (5%) was the cleavage of caspase-3 evidenced (Figure 3).

On the other hand, the poly (ADP-ribose) polymerase (PARP) family are proteins responsible for DNA repair [30]. PARP-1 perceives fractures in DNA and contributes to certain processes that lead to its reparation [31]. Caspase-3 activity is the main enzyme responsible for the cleavage and activation of PARP-1 when cell death is initiated [32]. In accordance with the results observed in the evaluation of caspase-3 activity, the 5% DMSO dose showed the cleavage of PARP-1 (Figure 4).

Despite having no evidence of proapoptotic marker (caspase-3 or PARP-1) activity at lower concentrations, minor toxicity may still have occurred without showing signs in these tests. Nevertheless, these results helped us to determine a range to focus on during the antiproliferative assays.

### 2.2. Antiproliferative Assays

The antiproliferative influence of DMSO in RA FLSs was examined using 3-(4,5-dimethylthiazol-2-yl)-2,5-diphenyltetrazoliumbromide (MTT) assays. Looking for better precision, we narrowed the range of DMSO concentrations to 0, 0.01, 0.05, 0.1, 0.5, 1 and 5%. The five different RA FLSs (60F, 71F, 72F, 65H and 69P) were evaluated. The results showed no significant differences between the RA FLSs from the five patients, regardless of age or joint (Figure 5), after 24 h exposure to DMSO. Considering the convergence of the results (Table 2), concentrations lower than 0.01% of DMSO could be considered safe in RA FLSs. For 0.05% of DMSO, approximately 1–4% of toxicity was detected (Figure 5 and Table 2). Toxicity became significant at 0.1% DMSO, as it reached approximately 5–12%. Concentrations above 0.1% should not be used, as significant toxicity was observed (≈15%). As expected, at the 5% DMSO concentration, most of the cells were floating in the culture medium after 24 h of exposure to the solvent, suggesting apoptotic events and/or the degradation of cell membrane adhesion proteins [24,25].

The toxicity depending on the exposure time to DMSO (24, 48 and 72 h) was also evaluated in three samples (60F, 71F and 72F). In all cases, the cells of the three patients showed a similar behavior, and this was why only the results associated with 71F were presented here. Interestingly, the results showed how the exposure time to DMSO increased the toxicity considerably (Figure 6). While a 24 h exposure using 0.01% of DMSO, or even 0.05%, can be considered safe, a longer exposure started to show significant toxicity. This was more obvious at higher concentrations, where a longer exposure time was clearly tied to higher toxicity. Therefore, 72 h exposure is not recommended when DMSO is used in vitro on FLSs.

## 3. Conclusions

We demonstrated that DMSO can be used in human RA FLSs to a maximum of 0.05% for a 24 h exposure. For longer exposures, only 0.01% can be considered safe and nontoxic. In addition, the results suggested that DMSO toxicity does not depend on the origin of RA FLSs, since the results in the antiproliferative assays did not show significant differences between different patients and different joints. Overall, this study might help researchers working on human fibroblast-like synoviocytes to establish reliable protocols when DMSO is required for the preparation of stock solutions of a poorly water-soluble drug.

## 4. Materials and Methods

### 4.1. General

DMEM medium, fetal bovine serum (FBS), L-glutamine and penicillin–streptomycin were bought from Gibco BRL. 3-(4,5-Dimethylthiazol-2-yl)-2,5-diphenyltetrazoliumbromide (MTT), dimethyl sulfoxide (DMSO), collagenase and dispase II were purchased from Sigma-Aldrich. Human anti-β-actin was bought from Sigma Aldrich. Caspase-3 and PARP-1 were purchased from Cell Signaling Technologies (Ozyme, Saint-Cyr-l’École, France). All reagents used in this work were purchased from commercial sources and used without further treatment or purification.

### 4.2. Preparation of Human RA Fibroblast-like Synoviocytes

FLSs were extracted and isolated from fresh synovial biopsies obtained from five RA patients undergoing finger, hand and pelvis arthroplasties. All patients satisfied the 1987 American Rheumatism Association criteria for RA [33]. The mean age of the patients was 67.4 ± 4.9 years (range 60–72 years). The mean disease duration was 8.7 ± 2.3 years. At the moment of surgery, the disease activity score (DAS 28) was greater than 3.2. These scientific activities were approved by local institutional review boards, and all subjects gave written informed consent. Synovia were cut into little pieces and digested with 1.5 mg/mL collagenase–dispase for 4 h at 37 °C, as previously described [34]. After centrifugation, cells were resuspended in DMEM supplemented with 10% FCS, 4.5 g/L L-glutamine, 100 U/mL penicillin and 100 μg/mL streptomycin (Gibco BRL) at 37 °C in a humidified atmosphere containing 5% (*v*/*v*) CO_2_. Subsequently, after 48 h, nonadherent cells were washed using phosphate-buffered saline (PBS). Adherent cells (macrophage-like and RA FLSs) were cultured in a complete DMEM medium, and, at confluence, cells were trypsinized and only RA FLSs were passed. These cells were used between passages 4 and 8, when they morphologically resembled FLSs after indirect immunofluorescence study (see subsection Culture of Human RA FLSs and Treatment). RA FLSs were cultured for 45–60 days before experimentation. This interval granted the exclusion of all possible interactions resulting from any preoperative treatment (with nonsteroidal anti-inflammatory drugs, analgesics, steroids or disease-modifying antirheumatic drugs).

### 4.3. Culture of Human RA FLSs and Treatment

Between passages 4 and 8, RA FLSs underwent a dissociation process using trypsin. Cell count and viability were determined, and cells were plated in culture plates or flasks (Falcon, Dutscher SA, Bernolsheim, France). Viability was always greater than 95% and was measured with histological staining and exclusion using trypan blue [35], at the start and at the end of culture.

### 4.4. Flow Cytometry

In total, 2 × 10^5^ RA FLSs were cultured in DMEM in a 25 cm^2^ flask and incubated at 37 °C and 5% CO_2_ for 24 h. Then, the desired percentage (*v*/*v*) of DMSO was added to the medium, carefully homogenized and the cells were incubated at the same conditions described before. After 24 h, RA FLS cells were trypsinized and added to 200 µL of PBS. In total, 5 µL of propidium iodide (PI) was added as internal standard. Flow cytometry was performed using a BD FACSCalibur flow cytometer.

### 4.5. Protein Extraction and Western Blot Assays

For total protein extraction, the RA FLSs were washed in 1 mL of PBS; then, the total cell pool was centrifuged at 4 °C at 200× *g* for 5 min. Next, it was homogenized in a radioimmunoprecipitation (RIPA) lysis buffer (50 mM HEPES ((4-(2-hydroxyethyl)-1-piperazineethanesulfonic acid)), pH 7.5, 150 mM NaCl, 1% sodium deoxycholate, 1% NP-40, 0.1% SDS, 20 mg/mL of aprotinin) containing protease inhibitors (Complete^TM^ Mini, Roche Diagnostics, Paris, France) according to the manufacturer’s instructions. Proteins (60 µg) were separated with electrophoresis on 10% SDS–PAGE gels, transferred to polyvinylidene fluoride (PVDF) membranes (Amersham Pharmacia Biotech, Amersham, United Kingdom) and probed with human caspase-3 (mouse) or human PARP-1 (rabbit) antibodies. After incubation, secondary antibody blots were developed using the ECL Plus Western Blotting Detection System (Amersham Pharmacia Biotech, Amersham, United Kingdom) and G: BOX system (Syngene, Ozyme, Saint-Cyr-l’École, France). Membranes were then reblotted with human anti-β-actin (Sigma Aldrich, St. Quentin Fallavier, France) used as a loading control.

### 4.6. Antiproliferative Assays

RA FLSs were trypsinized in a DMEM culture medium. Homogeneous solutions were prepared in 10 mL of medium with 7 × 10^5^ cells. In a 96-well plate, 7000 cells per well (100 µL of the solution) were poured and the cells were incubated at 37 °C and 5% CO_2_. After 24 h of incubation, 100 µL of DMEM medium, with the desired concentration of DMSO (0, 0.01, 0.05, 0.1, 0.5, 1 and 5%,) was poured per row into the 96-well plate and then incubated for 24, 48 or 72 h in the same conditions described above. After this time, 10 µL of MTT solution (5 g/L) was added per well and the plate was then put again inside the incubator for 4 h. Following this period, the medium was removed, 200 µL of DMSO were added per well, and the plate was stirred softly for 3 min. Absorbance after the MTT assay was measured at 540 nm using a Dynex Triad Multi Mode Microplate Reader, Dynex Technologies (Chantilly, VA, USA). The assays were carried out three times.

### 4.7. Statistical Analysis

All quantitative results were expressed as the mean ± 3 standard deviation (SEM) of separate experiments using Excel (Microsoft Office, Version 2019, Microsoft Corporation, Washington, DC, USA). Data normalization was carried out separately between the cells of each patient, corresponding to 0% in each lineage. Statistical significance was evaluated with the two-tailed unpaired Student’s *t*-test, *p*-value < 0.001 (***) and < 0.05 (*).

## Figures and Tables

**Figure 1 molecules-27-04472-f001:**
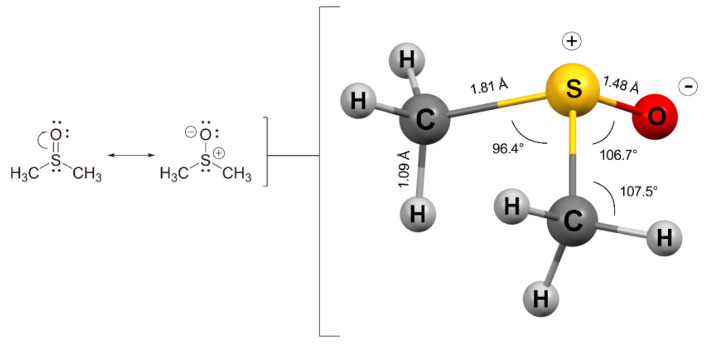
Formulation of the DMSO structure through a resonance hybrid (**left**) of the S=O double bond and the structure with a polarized S–O bond (**right**). This representation was more in agreement with the physical properties and reactivity observed for DMSO [22]. The angles and distances corresponded to the structure with a polarized S–O bond.

**Figure 2 molecules-27-04472-f002:**
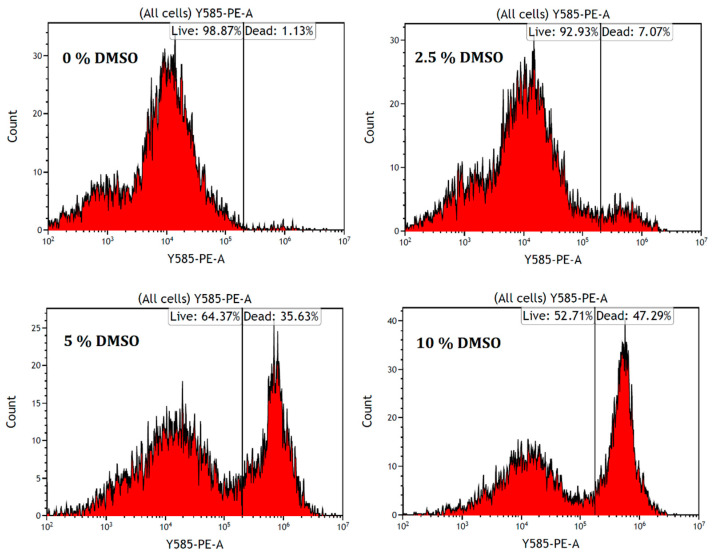
Graphic representation of the population of alive/dead cells determined with flow cytometry. RA FLSs were cultured and treated as described in the experimental part. Propidium iodide (PI) was used as the internal standard. No significant difference was observed with 7.5% (data not shown) and 10% DMSO.

**Figure 3 molecules-27-04472-f003:**
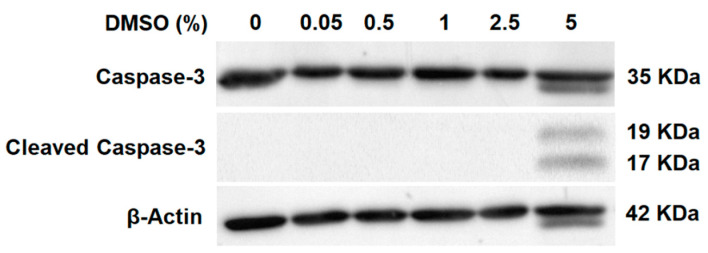
Identification of the presence of caspase-3 and cleaved caspase-3 using Western blot. ß-actin was used as a loading control. In total, 2 × 10^6^ RA FLSs were seeded in a 75 cm^2^ flask and incubated 24 h at 37 °C and 5% of CO_2_. After 24 h, DMSO was added and the medium homogenized gently. After 24 h, cells were trypsinized and washed with PBS. Protein extraction and Western blot were performed as described in the experimental part. All experiments were conducted in triplicate.

**Figure 4 molecules-27-04472-f004:**
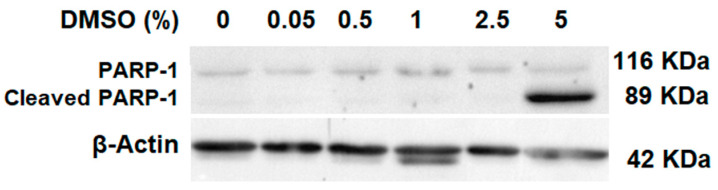
Identification of the presence of PARP-1 and cleavage of PARP-1 with Western blot. ß-actin was used as a loading control. Same method as described for caspase-3.

**Figure 5 molecules-27-04472-f005:**
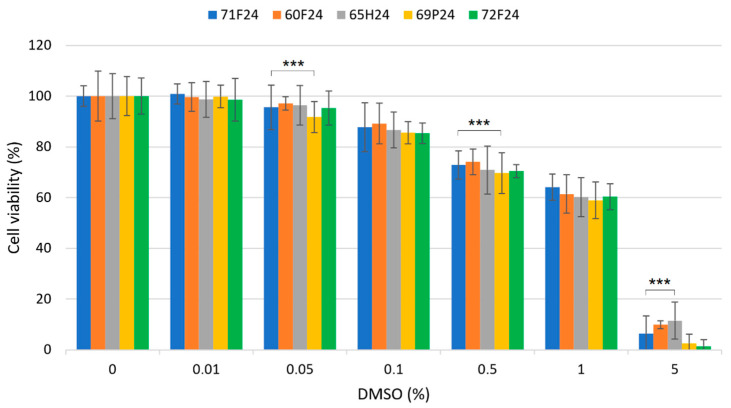
Results of the MTT assays in RA FLSs from the five patients after 24 h of exposure at different DMSO concentrations. Antiproliferative assays were performed as described in the experimental part. Statistical significance was evaluated with two-tailed unpaired Student’s *t*-test, *p*-value < 0.001 (***).

**Figure 6 molecules-27-04472-f006:**
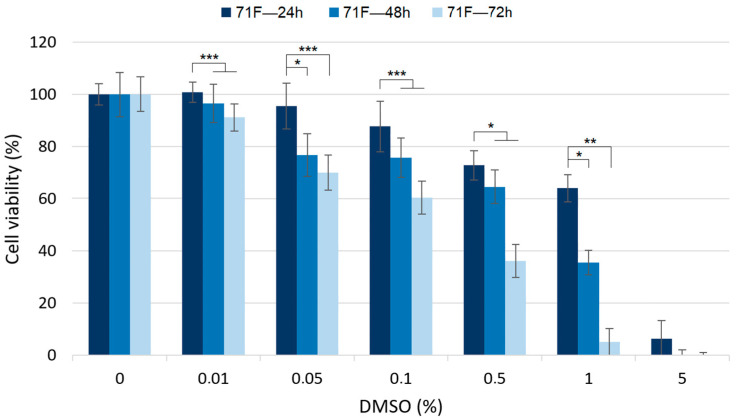
Results of the MTT assays in 71F RA FLSs after 24, 48 and 72 h of exposure at different DMSO concentrations. Assays were carried out as described in the experimental part. Statistical significance was evaluated with two-tailed unpaired Student’s *t*-test, *p*-value < 0.001 (***), < 0.01 (**) and < 0.05 (*).

**Table 1 molecules-27-04472-t001:** Standard physicochemical characteristics of water (left) and DMSO (right) [21,22,23].

	Water	DMSO
Chemical formula	H_2_O	C_2_H_6_OS
Appearance	Colorless liquid	Colorless liquid
Molecular weight	18.02	78.13
Polarity	1.000	0.444
Boiling point	100.0	189.0
Melting point	0.0	17.9
Density (g/mL)	0.9982	1.1010
Viscosity (cP) at 20 °C	1.002	1.996
Dipole moment (Debye)	1.850	3.960
Ebullioscopy constant (K·kg/mol)	0.513	3.220
Cryoscopic constant (K·kg/mol)	1.860	3.850
Specific heat at 25 °C (cal/g)	1.000	0.470
pK_a_ (25 °C)	14.0	35.1
Flash point (open dish, °C)	n/a	95
Dielectric constant at 20 °C	80.4	48.9
Refractive index	1.4793	1.333
Point group	C2v	Cs

**Table 2 molecules-27-04472-t002:** Results of the MTT assays carried out as described in the experimental part. The cell viability percentage is expressed as the main of three assays ± 3 sigma standard deviation.

Entry	DMSO (%)	60F	65H	69P	71F	72F	Average (%)
1	0	100.0 ± 9.7	100.0 ± 8.8	100.0 ± 7.8	100.0 ± 4.1	100.0 ± 7.1	100.0 ± 7.6
2	0.01	99.6 ± 5.7	98.7 ± 7.1	99.8 ± 4.5	100.9 ± 3.9	98.6 ± 8.4	99.7 ± 5.9
3	0.05	97.1 ± 2.7	96.4 ± 7.8	91.8 ± 6.1	95.6 ± 8.7	95.4 ± 6.7	95.5 ± 6.4
4	0.1	89.2 ± 8.0	86.7 ± 7.1	85.6 ± 4.4	87.7 ± 9.6	85.4 ± 4.0	86.6 ± 6.6
5	0.5	74.1 ± 5.1	70.9 ± 9.4	69.6 ± 8.0	72.9 ± 5.6	70.5 ± 2.5	71.7 ± 6.1
6	1	61.4 ± 7.5	60.2 ± 7.7	58.9 ± 7.3	64.1 ± 5.2	60.4 ± 5.1	62.2 ± 6.6
7	5	9.8 ± 1.6	11.5 ± 7.3	2.5 ± 3.5	6.4 ± 7.0	1.4 ± 2.5	3.9 ± 4.4

## Data Availability

Not applicable.

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
