# Peer review of "Dimethyl Sulfoxide: A Bio-Friendly or Bio-Hazard Chemical? The Effect of DMSO in Human Fibroblast-like Synoviocytes"

_molecules, 2022, doi:10.3390/molecules27144472_

Round 1

Reviewer 1 Report

In this manuscript, Liagre and coworkers evaluated toxicity effects of DMSO in human derived fibroblast-like synoviocytes (FLS). The authors quantified DMSO induced toxicity by cell adhesion phenotype, propidium iodide staining, caspase-3 and PARP cleavage. The authors further studied the cell viability under different DMSO concentrations with different incubation time by MTT assays. This work is significant in systematically evaluating DMSO toxicity in the tissue specific FLS cell lineage from human. The data in this manuscript are straightforward and well documented. Few minor statistics details are needed before publication in Molecules.

In Table 2, Figure 5 and 6, how are the MTT data normalized? Are the data cross-normalized between different cell lines to a 0% DMSO group from a specific cell line or separately normalized to corresponding 0% groups in each lineage? This statistical detail should be provided to readers. 

In Figure 5, the authors showed significant intra-group statistical differences (p<0.001 between 5 cell lines in each group), which is not correct based on the data. The same statistics issue also happens in Figure 6, 0 and 0.01 groups. It should be clearly stated that which two bars in these figures are compared for each illustration in the figures. 

Author Response

Reviewer 1

Comments and Suggestions for Authors

In this manuscript, Liagre and coworkers evaluated toxicity effects of DMSO in human derived fibroblast-like synoviocytes (FLS). The authors quantified DMSO induced toxicity by cell adhesion phenotype, propidium iodide staining, caspase-3 and PARP cleavage. The authors further studied the cell viability under different DMSO concentrations with different incubation time by MTT assays. This work is significant in systematically evaluating DMSO toxicity in the tissue specific FLS cell lineage from human. The data in this manuscript are straightforward and well documented. Few minor statistics details are needed before publication in Molecules.

In Table 2, Figure 5 and 6, how are the MTT data normalized? Are the data cross-normalized between different cell lines to a 0% DMSO group from a specific cell line or separately normalized to corresponding 0% groups in each lineage? This statistical detail should be provided to readers. 

First of all, we thank Reviewer 1 for the feedback. Regarding the normalization of the data, it has been carried out separately between the cells of each patient, normalizing each one of them to the corresponding 0% in each lineage. To clarify this aspect, in section “4.7 Statistical analysis” (page 8, line 224), has been added: "Data normalization has been carried out separately between the cells of each patient, corresponding 0% in each lineage".

In Figure 5, the authors showed significant intra-group statistical differences (p<0.001 between 5 cell lines in each group), which is not correct based on the data. The same statistics issue also happens in Figure 6, 0 and 0.01 groups. It should be clearly stated that which two bars in these figures are compared for each illustration in the figures. 

We agree with Reviewer 1. The way of displaying the P-value was not clear or intuitive. All values have been calculated again and Figures 5 and Figure 6 have been changed. P-values are now only shown in those significant parts.

Reviewer 2 Report

The MS entitled "Dimethyl sulfoxide: A bio-friendly or bio-hazard chemicals? The effect of DMSO in human fibroblast-like synoviocytes” is well written and presents a substantial amount of very interesting data.

1.      Keywords: Is it necessary all of them! Please check based on mesh term.

2.      Insert a ref. for each section of western blot and real time PCR formula.

3.      Write a title for each figure legend and it is not necessary to duplicate method in legends, therefore revise them.

4.      In the section of results: It is better to mention p-value in significant parts.

5.      Conclusion: this study should help researchers working…. it is exaggerated, revise it.

Author Response

Reviewer 2

Comments and Suggestions for Authors

The MS entitled "Dimethyl sulfoxide: A bio-friendly or bio-hazard chemicals? The effect of DMSO in human fibroblast-like synoviocytes” is well written and presents a substantial amount of very interesting data.

  1. KeywordsIs it necessary all of them! Please check based on mesh term.

We would like to thank Reviewer 2 for the comments and feedback. As for keyboards, we have removed “toxicity, MTT assays, caspase-3 and poly (ADP-ribose) polymerase”. In this way, now the script shows the following keyboards: fibroblast-like synoviocyte, dimethyl sulfoxide and rheumatoid arthritis.

  1. Insert a ref. for each section of western blot and real time PCR formula.

I am really sorry but I don't understand this remark: in your article no real time PCR or even semi-quantitative RT-PCR has been performed...Only western blots for protein expression have been performed.

On the other hand, I don't understand what he means when he says "Insert a ref. for each section of western blot"?

  1. Write a title for each figure legend and it is not necessary to duplicate method in legends, therefore revise them.

We agree with Reviewer 2 on this point, so in Figure 4 “2x106 RA FLS were seeded in [...] All experiments were done in triplicate” has been removed and replaced with “Same method as described for caspase -3".

  1. In the section of results: It is better to mention p-value in significant parts.

We agree with Reviewer 2, Figure 5 and Figure 6 have been changed, to show the P-value only in the significant parts. In addition, all values have been checked again.

  1. Conclusion:this study should help researchers working…. it is exaggerated, revise it.

Page 6, line 157 – Replaced the word “should” with “might”.

Page 6, line 158 – The word “reliable” has been removed.
